# Phytohormone Production Profiles in *Trichoderma* Species and Their Relationship to Wheat Plant Responses to Water Stress

**DOI:** 10.3390/pathogens10080991

**Published:** 2021-08-06

**Authors:** María Illescas, Alberto Pedrero-Méndez, Marcieli Pitorini-Bovolini, Rosa Hermosa, Enrique Monte

**Affiliations:** Institute for Agribiotechnology Research (CIALE), Department of Microbiology and Genetics, University of Salamanca, Campus de Villamayor, C/Duero, 12, 37185 Salamanca, Spain; millesmor@usal.es (M.I.); alberto.pedrerom@usal.es (A.P.-M.); m.bovolini@hotmail.com (M.P.-B.); rhp@usal.es (R.H.)

**Keywords:** fungal phytohormones, gibberellin, auxin, cytokinin, ACC deaminase, drought

## Abstract

The production of eight phytohormones by *Trichoderma* species is described, as well as the 1-aminocyclopropane-1-carboxylic acid (ACC) deaminase (ACCD) activity, which diverts the ethylene biosynthetic pathway in plants. The use of the *Trichoderma* strains *T. virens* T49, *T. longibrachiatum* T68, *T. spirale* T75 and *T. harzianum* T115 served to demonstrate the diverse production of the phytohormones gibberellins (GA) GA_1_ and GA_4_, abscisic acid (ABA), salicylic acid (SA), auxin (indole-3-acetic acid: IAA) and the cytokinins (CK) dihydrozeatin (DHZ), isopenteniladenine (iP) and trans-zeatin (tZ) in this genus. Such production is dependent on strain and/or culture medium. These four strains showed different degrees of wheat root colonization. Fresh and dry weights, conductance, H_2_O_2_ content and antioxidant activities such as superoxide dismutase, peroxidase and catalase were analyzed, under optimal irrigation and water stress conditions, on 30-days-old wheat plants treated with four-day-old *Trichoderma* cultures, obtained from potato dextrose broth (PDB) and PDB-tryptophan (Trp). The application of *Trichoderma* PDB cultures to wheat plants could be linked to the plants’ ability to adapt the antioxidant machinery and to tolerate water stress. Plants treated with PDB cultures of T49 and T115 had the significantly highest weights under water stress. Compared to controls, treatments with strains T68 and T75, with constrained GA_1_ and GA_4_ production, resulted in smaller plants regardless of fungal growth medium and irrigation regime.

## 1. Introduction

The establishment of microbial symbioses to promote plant growth and nutrient acquisition by beneficial microbes have been correlated to the biosynthesis of plant growth regulators and phytohormones [1,2]. It is well established that, in addition to inducing host hormone synthesis, pathogenic and symbiotic fungi can also modulate the hormonal network of plants, as they produce by themselves small amounts of phytohormones to serve their purpose. Jasmonic acid (JA), auxin (indole-3-acetic acid: IAA), cytokinins (CK), gibberellins (GA), ethylene (ET), abscisic acid (ABA) and salicylic acid (SA) of fungal origin are involved in favoring tissue colonization and nutrient uptake, by means of plant development control and activation of signaling events during biotic and abiotic stresses [3]. Thus, auxin and GA producing endophytic fungi can enhance host plant growth and alleviate adverse effects of an abiotic stress, opening up the possibility of their use to improve agricultural productivity under adverse soil conditions [4]. In the same case is *Trichoderma*, a fungal biocontrol agent that includes species that are well known for their ability to produce fungal and oomycete cell wall degrading enzymes [5], scavenging reactive oxygen species (ROS) and causing plant cell wall hydrolysis [6,7] to facilitate the endophytic colonization of root tissues in competition with pathogens [8]. Selected *Trichoderma* species also produce effector molecules capable of triggering signaling cascades in the plant [9,10,11] that lead to the induction of systemic resistance to biotic and abiotic stresses as well as growth promotion [12,13]. In this regard, rhizosphere competent species have evolved to manipulate root development, plant immunity and stress tolerance by producing phytohormones [14]. It has been shown that *T. atroviride*, *T. virens* and *T. harzianum* produce IAA, *T. parareesei* produces SA and *Trichoderma* sp. produces IAA and GA without any inducers, although it is known that their production levels depend on the amount of tryptophan (Trp) present in the medium [15,16,17,18,19,20,21]. *T. asperellum* also releases ABA together with IAA and GA into the culture medium, and its application to cucumber promoted seedling growth and alleviated the effects of salt stress [22]. The production of IAA by *T. harzianum* has been related to the biocontrol of anthracnose disease and improved growth of sorghum plants [21]. The application of *T. parareesei* T6 or *T. harzianum* T34 to tomato seeds also improved the tolerance of plants to salt stress and enhanced the growth when plants grew under this adverse condition [23,24]. *T. afroharzianum* (formerly *T. harzianum*) T22 improved tolerance of tomato seedlings to water deficit [25]. The colonization of cocoa seedlings by *T. hamatum* DIS 219b enhanced seedling growth, altered gene expression, and delayed the onset of the cocoa drought response in leaves [26]. Similarly, *T. atroviride* ID20G inoculation of seeds ameliorated drought stress-induced damages by improving antioxidant defense in maize seedlings [27]. The same happened with the improved drought tolerance observed in rice genotypes inoculated with *T. harzianum* Th-56, in which the antioxidant machinery was activated in a dose-dependent manner [28].

IAA is the phytohormone that regulates the plant’s development of the primary and lateral roots [29]. It has been described in other fungi such as *Serendipita indica* that plant IAA levels have little or no effect on the beneficial fungus-mediated growth promotion, as the plant is very sensitive to changes in IAA concentration and a slight increase in this phytohormone, rather than stimulate, can limit growth [30]. It is well known that additional *Trichoderma* metabolites and proteins are involved in the regulation of IAA signals in the plant, leading to root hair growth and increased root mass development [31,32,33]. This evidence seems to indicate that rather than a major function in root morphogenesis, IAA and the other phytohormones of fungal origin play a role in interconnecting plant development and defense responses as a component of the complex *Trichoderma*-regulated phytohormone networking in plants [12,13].

To further complicate the understanding of this issue, ethylene (ET) is a phytohormone which regulates plant growth, development, and senescence, and it is well established that low ET concentrations in the root zone correspond to higher shoot growth [34]; therefore, limiting the levels of ET serves to increase agricultural production. A strategy followed by many rhizospheric microorganisms to favor plants consists of reducing the concentration of 1-aminocyclopropane-1-carboxylic acid (ACC), the precursor molecule of ET, by means of the ability to produce the enzyme ACC deaminase (ACCD). *Trichoderma* strains have the capacity to produce ACCD. This is the case of *T. longibrachiatum* TL-6, involved in promoting wheat growth and enhancing plant tolerance to salt stress [35], and *T. asperelloides* (formerly *T. asperellum*) T203 that by being able to regulate the endogenous ACC levels stimulates root elongation of cucumber [36] and *T. asperellum* MAP1, which enhanced wheat plant tolerance to waterlogging stress [37].

Wheat is one of the most important crops in the world, providing one-fifth of proteins and calories in human diet, and its extensive production is often subjected to non-irrigation conditions [38]. ROS are key players in the complex signaling network of plant responses to drought stress, so it is essential to maintain ROS at non-toxic levels in a delicate balancing act between ROS production, involving ROS generating enzymes and the unavoidable production of ROS during basic cellular metabolism, and ROS-scavenging pathways [39]. The application of *Trichoderma* to wheat triggers systemic defense pathways [40] and seems to be a good choice to minimize damage caused by abiotic stresses [35,37], also limiting environmental pollution. There is sufficient evidence to consider that *Trichoderma* association can help plants in sustaining drought stress by increasing: (i) the expression of antioxidative enzymes that alleviate the damage caused by the accumulation of ROS and modulating the balance of plant’s phytohormones [25,41]; (ii) the absorption surface that leads the plant to improve water-use efficiency [33]; and (iii) the synthesis of phytohormones and phytohormonal analogues to promote plant performance.

In the present work, we have used four *Trichoderma* strains of four different species representing the genetic diversity of the genus, in which we analyzed their capacity for wheat root colonization, measured ACCD activity, and production levels of the phytohormones GA_1_, GA_4_, ABA, SA, IAA and the CK dihydrozeatin (DHZ), isopenteniladenine (iP) and trans-zeatin (tZ) in medium supplemented or not with Trp. We then analyzed the ability of PDB and PDB-Trp cultures of these strains to favor wheat plants in their growth and their adaptation to grow under water stress. In addition, activities related to the reduction of ROS levels in plants were measured as an indication of good performance of plants inoculated with *Trichoderma* strains.

## 2. Results

### 2.1. Molecular Characterization of Trichoderma Strains

The identity of the four soil-isolated *Trichoderma* strains used in this study was confirmed at the species level by analysis of the sequences of ITS1-ITS4 region and a fragment ca. 600 bp in length of *tef1**α* gene. They had sequences identical to those of ex-type strains or representative species available in databases. They were identified as: *T. virens* T49, *T. longibrachiatum* T68, *T. spirale* T75 and *T. harzianum* T115, and the accession numbers of their sequences in the GenBank are shown in Table 1. These strains showed significant differences in growth and degree of sporulation after culturing in three different culture media (Table 2). Strain T49 showed the highest growth rate when cultivated on PDA, PDA-Trp and MEA while T75 was the lowest growing on these media. The growth differences observed for T68 between PDA and PDA-Trp indicate that the addition of Trp negatively affected the growth of this strain. The effect of culture medium was also observed on the sporulation degree, with T75 being the strain that significantly showed the lowest values on PDA or PDA-Trp, and T49 the highest on MEA.

### 2.2. Differences in Colonization of Roots of Wheat Seedlings by Trichoderma Strains

In order to perform a comparative analysis of the wheat root colonization ability among the four *Trichoderma* strains, we determined the proportion of fungal DNA vs. plant DNA from qPCR data in 10-day-old seedling roots at 42 h after fungal inoculation. As shown in Table 3, strains T49, T75 and T115 colonized the roots, with the highest rates for T49 and T75 (*p* < 0.05), while T68 showed no colonization.

### 2.3. Differences in ACCD Activity and Phytohormonal Profiles in Trichoderma Strains

The ACCD activity was calculated for all four strains after growing them for four days in synthetic minimal medium. Strain T115 showed significantly higher specific ACCD activity (1.8 mmol of α-ketobutyrate per mg of protein) compared to that of the other three strains (0.09 to 0.20 mmol α-ketobutyrate per mg of protein) (Tukey test at *p* < 0.05), which showed no significant differences between them.

The production of eight phytohormones by the four *Trichoderma* strains was measured in PDB medium with and without Trp. Since the PDB medium is composed of plant material, uninoculated media were used as controls. Under these two conditions, a comparative analysis of the production profiles of GA_4_, GA_1_, ABA, SA, IAA, DHZ, iP and tZ by strains T49, T68, T75 and T115 is shown in Figure 1. When compared to the control conditions of each culture medium in a one-way ANOVA, not all *Trichoderma* strains exhibited production of the eight phytohormones in both media. There was an effect of the variable “strain” (*p* < 0.001) and variable “medium” (*p* < 0.001), and their combination on the production of seven of the phytohormones investigated, according to a two-way ANOVA (*p* < 0.001).

Particularly, the CK iP was the only one that showed no significant effect for the combination of the two variables. *T. virens* T49 significantly exhibited the highest levels of GA_4_ in both media, being much higher in PDB-Trp (*p* < 0.001). Considering the phytohormone production profiles as a whole *T. longibrachiatum* T68 did not stand out for any of them. In addition, GA_4_ levels were lower for this strain than those detected in its controls, which would be indicative of the metabolization of this molecule present in the medium. Similar behavior was observed only for GA_1_ with strains T75 and T115 in PDB-Trp medium. *T. spirale* T75 showed the highest production levels of SA, IAA and CK. The biosynthesis of SA and CK by this strain did not respond to the addition of Trp to the culture medium. However, strain T75 in PDB-Trp increased IAA levels by about 80 times. On the contrary, strain T49 showed higher levels of IAA production in PDB than in PDB-Trp. *T. harzianum* T115 was the strain in which the levels of GA_1_ and ABA production in PDB were significantly the highest.

### 2.4. The Effect of Trichoderma Strains on Wheat Plants under Drought Stress

Greenhouse-grown wheat plants were used to evaluate the effect of PDB and PDB-Trp cultures of the four *Trichoderma* strains when they were applied to the plant substrate. Plant fresh and dry weight and conductance parameters were measured after 30 days of growth under optimal irrigation and 1/3 of the watering applied during the third and fourth weeks (water stress) (Table 4 and Table 5). Representative phenotypes observed in wheat plants treated with the different *Trichoderma* cultures and irrigation regimes are shown in Figure 2. In a broad sense, the one-way ANOVA results showed the existence of significance for the factors “strain” (*p* < 0.001), “culture medium” (*p* < 0.001) and “stress” (*p* < 0.001). Two different plant responses were observed for *Trichoderma* cultures from both PDB and PDB-Trp media. Therefore, plants treated with T68 and T75 PDB cultures significantly showed the lowest fresh and dry weight compared to the other treatments under optimal irrigation conditions (Table 4). On the other hand, under water stress conditions, plants treated with PDB cultures of T49 and T115 had significantly the highest weights, with an increase of ca. 100%. Regarding conductance values, wheat plants showed significantly higher numbers with T49 and T115 PDB cultures under optimal irrigation conditions, whereas the control presented a significant reduction compared to any of the four *Trichoderma* strains applied under water stress.

In a similar way, wheat plants treated with T68 and T75 PDB-Trp cultures had significantly lower fresh and dry weight values than control plants or those treated with T49 and T115 PDB-Trp cultures under optimal irrigation conditions (Table 5). However, no differences in weight and conductance values under water stress were observed among treatments with the sole exception of those plants treated with T68 or T75 PDB-Trp cultures, which gave significantly lower dry weight values (Table 5). A two-way ANOVA for dry weight data showed significance for “culture medium” × “stress” (*p* < 0.05); and for conductance data, all combinations (“strain” × “culture medium”, “strain” × “stress”, “culture medium” × “stress”; *p* < 0.001) were significant. Additionally, a three-way ANOVA showed significance for the combination “strain” × “culture medium” × “stress” for fresh and dry weight (*p* < 0.05) and for conductance (*p* < 0.01).

Endogenous H_2_O_2_ content in wheat leaf from 30-day-old plants did not show variation in unstressed plants, neither in the control nor with *Trichoderma* regardless of the presence of Trp in the medium to grow the fungus (Figure 3). Water stress control plants from the PDB condition showed a significant increase in H_2_O_2_ content compared to those challenged with *Trichoderma*. However, PDB-Trp condition stressed control plants showed lower levels of H_2_O_2_ than plants treated with *Trichoderma* cultures, which in turn were significantly different, with the highest levels for the T115 treatment. The two-way ANOVA showed significance of the three considered factors and their pairwise combinations (*p* < 0.01) with the only exception of “culture medium” × “stress”, while the three-way ANOVA was significant for the three factors together (*p* < 0.01).

The values calculated for three antioxidant enzymes in wheat plants are shown in Figure 4. Compared to the respective controls, in the absence of water stress and when Trp was not added to the fungal culture medium, the application of *Trichoderma* cultures resulted in significant SOD activity increase, except for the T115 treatment. *Trichoderma* application significantly decreased POD without changing CAT activity. Unstressed plants treated with *Trichoderma* PDB-Trp cultures increased SOD activity compared to the control. However, POD activity only decreased significantly in T115-treated plants, with CAT activity being lower than that of the control in all cases. Differences were also observed among *Trichoderma* treatments as the decrease in CAT activity was significantly lower in plants challenged with T68. Under the condition of water stress, no significant differences were detected in SOD, POD and CAT activities of plants subjected to any of the PDB-Trp treatments compared to the control. However, in absence of Trp in *Trichoderma* cultures, the stressed plants responded to *Trichoderma* by lowering POD and CAT activities, and only strain T49 and T68 were able to significantly rise SOD activity. ANOVA values indicate that the factor “strain” had significance in the three tested enzymatic activities of plants (*p* < 0.001), while the factor “stress” had significance in SOD (*p* < 0.001) and CAT (*p* < 0.05), and factor “culture medium” in SOD (*p* < 0.001) and POD (*p* < 0.05). 

## 3. Discussion

*Trichoderma* is a very complex fungal genus that includes nearly 400 species [42]. The practical application of *Trichoderma* needs a correct molecular characterization as the biocontrol, biostimulation and other beneficial effects to plants should not be considered in broad terms, but at the level of strain. We have included in our study four strains belonging to four phylogenetically distant species to explore their behavior regarding how they promote growth and favor water-stressed wheat plants. Modern *Trichoderma* taxonomy suggests the analysis of three DNA barcodes (ITS, *tef1* and *rpb2*) [42], and we have achieved unambiguous species identification by ITS1-ITS4 and 600 bp in length of *tef1**α* gene sequencing. Two out of four strains identified belong to *T. harzianum* and *T. virens*, two species widely used as biocontrol agents in commercial practice [43,44]. The other two strains belong to species less used in biological control, although there is recent work on the efficacy of *T. spirale* and *T. longibrachiatum* in the control of plant pathogenic fungi [45,46]. 

Our study has been focused on the abilities of these strains to stimulate the growth of wheat plants and alleviate them from water stress. Root colonization ability is often a criterion for selecting *Trichoderma* strains beneficial to plants [12], and we found that wheat was not a host for strain T68. An important and little studied aspect of *Trichoderma* is the capacity to produce phytohormones that may be involved in plant interactions. Depending on the strain of *Trichoderma* and the composition of the culture medium, with or without addition of Trp, or the combination of both, the production of phytohormones was affected. The observed differences in phytohormone production could be affected by the degree of growth of the different strains. However, strain T68 showed good growth and sporulation performances on PDA and PDA-Trp and did not stand out in the production of any of the eight phytohormones tested in PDB and PDB-Trp. PDB has been used because it is a common medium for *Trichoderma* growth and because the production of IAA has been described in this medium supplemented with Trp [16]. As PDB contains molecules of plant origin, the uninoculated medium has been used as a control, with and without Trp addition, to subtract possible phytohormones already present in the fungal culture media. Although Trp-containing media seem to favor the production of IAA, this is not a rule, as strain T49 showed a behavior contrary to the other three *Trichoderma* strains. T49 and T115 were the only strains that produced GA_4_ and GA_1_, respectively, in medium not supplemented with Trp. However, the addition of Trp to the growth medium of the fungus induced GA_4_, but not GA_1_, production in both strains. Production of GA_3_ has been described in *T. harzianum*, and accumulation of this phytohormone in combination with IAA has been related to plant growth promotion [15,19]. The production of gibberellic acid by *Trichoderma* also cooperates with IAA and ACCD in the modulation of defense responses in wheat seedlings [18]. Production of GA_1_ and GA_4_ have been described in other fungi such as *Phoma*, *Penicillium* and *Aspergillus* as plant growth promoters under stress conditions [47,48]. In our case, we have seen that the production profiles of GA_1_ and GA_4_ are antagonistic, and in the strains that produce them, T49 and T115, their biosynthesis seems to be compensated. It is well known the antagonistic regulation of GA and ABA in plants [49], and this also occurs in *Trichoderma* for GA_4_ and ABA production. However, this was not the case of GA_1_, as strain T115 reached in PDB the highest levels of this phytohormone and ABA simultaneously. Regarding CK, it has been described that their production in fungi is related to hyphal growth and branching, and their accumulation allows better adaptation to stress and colonization of the roots, although the effect on fungal growth is made in a dose-dependent manner [3]. *T. spirale* T75 produced the highest amount of the three CK analyzed, DHZ, iP and tZ, in the two media used and this was accompanied by the slowest significant growth on PDA and PDA-Trp. As seen in plants [1], this strain showed the typical IAA-CK antagonism when cultured in PDB. However, strain T75 showed the highest IAA and CK production values in PDB-Trp. It should be noted that the production of IAA by strain T75 in PDB-Trp was particularly high and that the trend in all strains was that the addition of Trp reduced the CK levels. 

*Trichoderma* can manipulate the phytohormone regulatory network decreasing the ET precursor ACC through the ACCD activity [12,36]. The four *Trichoderma* strains exhibited ACCD activity although strain T115 showed 20 times more activity than the other three under identical growth conditions in a synthetic medium. These results are also a consequence of working with strains from genetically very distant species, given the great diversity that exists within the *Trichoderma* genus [50]. *Trichoderma* ACCD has also been described as a mechanism in enhancing wheat tolerance to salt stress [35] and waterlogging [37]. Our study has included the application of *Trichoderma* to wheat plants to analyze the effect on growth and tolerance to water stress. The greenhouse assay was conducted using mycelium plus culture supernatant of *Trichoderma* to inoculate the substrate where wheat plants were grown, and it is therefore difficult to assess the role of *Trichoderma* phytohormones in wheat plant responses. Under optimal irrigation conditions, none of the treatments with *Trichoderma* appeared to promote the growth of wheat plants. Moreover, two of the strains, T68 and T75, performed worse than the PDB and PDB-Trp control plants (Figure 2). Perhaps the smaller size and weight of plants compared to their controls (Table 4 and Table 5) may be because these two *Trichoderma* strains show no GA_1_ and GA_4_ production. It is noteworthy that strain T75, which produced as indicated above the highest concentrations of IAA in PDB-Trp, did not promote plant growth, which would indicate that fungal IAA contributes to the total concentrations of this phytohormone, but it is not the major player in root development as plant IAA does. The high levels of SA and CK reached by this strain (Figure 2) could be the cause of the phenotype observed in T75-treated plants. Since strain T68 was unable to colonize the wheat root, it may be releasing some other metabolites that could limit the growth of the plant. PDB cultures from strains T49 and T115, those producing maximum amounts of GA_4_, and GA_1_ and ACCD activity, respectively, were the ones that best increased plant tolerance to water stress, also being the ones that provided higher conductance and weight values in plants (Table 4). The importance of selecting a suitable strain of *Trichoderma* is a key point in this type of study, as it has been observed that the colonization of *Arabidopsis*, tomato and maize roots by *T. virens* Gv29.8 led to reduced growth of both roots and stems [7,51,52]. Nevertheless, plants treated with PDB cultures of strains T68 and T75 did not show increased growth but did show high conductance (Table 4) and a water stress tolerance phenotype compared with PDB control plants (Figure 2). The significant increases in conductance that we observed in plants from the *Trichoderma* PDB treatments compared to their control under water stress conditions agree with previous reports indicating that *Trichoderma* can ameliorate the conductance decline in drought stressed plants [26,53]. 

Plants treated with *Trichoderma* PDB cultures under water stress conditions significantly decreased the H_2_O_2_ content compared to the control, although no differences were detected under optimal irrigation condition. This result is in line with what has been described in maize treated with *T. atroviride* under drought stress [27]. In the present study, all *Trichoderma* strains were able to produce to a greater or lesser extent SA (Figure 2), this phytohormone being very important in the establishment of a plant oxidative burst in response to stress, but also in the upregulation of antioxidant metabolism [13]. The antioxidant level in plant was analyzed by measuring SOD, POD and CAT activities. In a broad sense and as expected, *Trichoderma* increased the SOD antioxidant activity of the plants under water stress conditions. These results would agree with those reported in stressed or infected tomato plants inoculated with *Trichoderma* [25,54]. Like in maize inoculated with *T. harzianum* under salt stress [55], we have also seen that *Trichoderma* application decreased POD and CAT activities under water deficit conditions. Considering the profiles observed for the three enzyme activities in wheat plants treated with *Trichoderma* PDB cultures, it seems that the effect of *Trichoderma* prevails over the stress condition in driving the plant’s antioxidant machinery. The addition of Trp to *Trichoderma* cultures did not appear to modify plant antioxidant enzyme profiles, upregulation of SOD and downregulation of CAT, under non-stressed conditions. However, stressed plants did not modify their antioxidant activity with respect to the control, and it seems that the Trp effect prevailed over the *Trichoderma* application. Finally, Trp is shown to play a prominent role in the response of wheat plants to water stress as the PDB-Trp control plants had higher weight and conductance values than the PDB control plants. The phenotype of PDB-Trp control plants agrees with the collapse observed in tomato plants over-stimulated with NPK fertilization and *Trichoderma* under salt stress [24]. However, the phenotypes of the *Trichoderma*-treated plants did not appear to be greatly affected by Trp supplementation.

The production of the phytohormones GAs, ABA, SA, IAA and CKs by *Trichoderma* species is a strain-specific characteristic and depends on the composition of the culture medium. These differences are a factor to be considered when exploring the beneficial effects of *Trichoderma* on plants. In this way, the *T. virens* T49 and *T. harzianum* T115 cultures were the best performers in alleviating wheat plants from water stress and it was precisely these two strains which exhibited GA_1_ and IAA, and GA_4_ and ABA production, respectively, in media not supplemented with Trp. The present work contributes to highlighting the role that the balance of phytohormone levels, to which *Trichoderma* contributes with its own production, plays in beneficial plant-*Trichoderma* interactions. In any case, the growth promotion and plant protection effects of *Trichoderma* are mechanisms with complex regulation that depends on other *Trichoderma* traits and not only on the production of phytohormones by this fungus. The results of this work are an example of the usefulness of *Trichoderma* strains in the protection of crop plants against abiotic stresses.

## 4. Materials and Methods

### 4.1. Trichoderma Strains

Four *Trichoderma* strains isolated from soil and representing different genotypes were used in this study: *T. virens* T49, *T. longibrachiatum* T68, *T. spirale* T75 and *T. harzianum* T115 (references of our collection, CIALE, University of Salamanca, Spain). Three out of four strains (T49, T68 and T75) have been included in a previous genetic diversity study and their ITS (internal transcribed spacer) 1 sequence was available [56]. Strains were routinely grown on potato dextrose agar (PDA, Difco Laboratories, Detroit, MI, USA) at 28 °C in the dark. For long-term storage, the strains were maintained at −80 °C in a 30% glycerol solution. 

#### 4.1.1. Assays of Trichoderma Growth and Sporulation 

For the determination of fungal growth, 5-mm-diameter PDA plugs of fungi were placed at the center of Petri dishes containing PDA, PDA-Trp or malt extract agar (MEA, Difco Laboratories Inc., Detroit, MI, USA) medium, plates were incubated at 28 °C in the dark, and colony diameters were recorded at two days. After 10 days of incubation at 28 °C, fungal spores were harvested and counted as previously described [23]. For each strain and medium, four replicates were performed.

#### 4.1.2. Molecular Characterization of Trichoderma Strains

DNA was obtained from mycelium collected from cultures in potato dextrose broth (PDB, Difco Laboratories Inc.) medium for 48 h as previously described [57]. The ITS regions of the nuclear rDNA gene cluster, including ITS1 and ITS2 and the 5.8S rDNA gene, and a fragment of the *tef1**α* gene were amplified with the primer pairs ITS1/ITS4 and EF1-728F/tef1rev, respectively, as described previously [56,58].

PCR products were electrophoresed on 1% agarose gels, the amplicons were excised from the agarose gels, and DNA purified and sequenced as previously described [58]. The sequences obtained were analyzed considering homology in the NCBI database with ex-type strains and taxonomically established isolates of *Trichoderma* as references. All sequences obtained in this study have been submitted to GenBank, and their accession numbers are indicated in Table 1.

#### 4.1.3. Root Colonization Assay

The quantification of *Trichoderma* DNA in wheat roots was performed by quantitative PCR (qPCR) as previously described [6,20], with some modifications. Wheat roots were collected from 10-day-old seedlings cultured in 10-mL flasks containing 8 mL of liquid Murashige and Skoog medium (MS, Duchefa Biochemie BV, Haarlem, Netherlands) supplemented with 1% sucrose, and inoculated with 10^5^ conidial germlings mL^−^^1^ of *Trichoderma* strain or not (control). Three seedlings per flask were used. *Trichoderma* germlings were obtained from 15 h cultures in PDB at 28 °C and 200 rpm. After 42 h of fungal inoculation, the wheat roots were collected, washed with sterile water, homogenized under liquid nitrogen, and kept at −20 ◦C until DNA obtainment. DNA was extracted using the Fast DNA Spin Kit for Soil (MP Biomedical LLC, Irvine, CA, USA). Four independent wheat-*Trichoderma* strain co-cultures were used for each fungal strain.

qPCR were performed with a Step One Plus thermocycler (Applied Biosystems, Foster City, CA, USA), using KAPA SYBR FAST (Biosystems, Buenos Aires, Argentine) and the previously described primer couples Act-F//Act-R (5′ATGGTATGGGTCAGAAGGA-3′ and 5′ ATGTCAACACGAGCAATGG) [6] and Act-Fw//Act-Rw (5′-TGACCGTATGAGCAAGGAG-3′//5′-CCAGACACTGTACTTCCTC-3′ [40], which amplify a fragment of the *actin* gene from *Trichoderma* and wheat, respectively. Reaction mixtures, prepared in triplicate with 1:10 diluted DNA, and PCR conditions were as previously describe [20]. Ct values were calculated and the amount of fungal DNA was estimated using standard curves; and finally values were normalized to the amount of wheat DNA in the samples. Each sample was tested in quadrupled.

#### 4.1.4. ACCD Activity of Trichoderma Strains

The ACCD activity of T49, T68, T75 and T115 strains was carried out as previously described [35,36] with some modifications. For each strain, 100 μL of spore suspension (1 × 10^6^ spores/mL) were inoculated in 10 mL of synthetic medium [59], and the cultures grown at 28 °C and 180 rpm for 4 days. The mycelia were collected, resuspended in 2.5 mL of Tris buffer 0.1 M (pH 8.5) and homogenized for 1 min. Toluene (25 μL) was added to a 200 μL aliquot and vortexed for 30 s, and 20 μL of 0.5 M ACC was added (Tris buffer was added in the control). The following steps, including the additions of HCl, 2,4-dinitrophenylhydrazine and NaOH, centrifugations, and the incubation periods of reactions, were as previously described [36]. ACC activity was analyzed quantitatively by measuring the amount of α-ketobutyrate produced by the deamination of ACC. α-ketobutyrate (10–200 μmol) was used for the standard curve and absorbance was measured at 540 nm. ACCD activity was expressed as mmol α-ketobutyrate mg^−1^ protein h^−1^. The Bradford protein assay was used to measure the protein total concentration in the samples [60] using the BioRad Promega Biotech Ibérica, Alcobendas, Madrid, Spain) reactive. Three independent replicate cultures were analyzed.

#### 4.1.5. Determination of Phytohormone-like Compounds by Trichoderma

The strains were grown in 200 mL of PDB and PDB with 200 mg/L of tryptophan (PDB-Trp) media at 28 °C and 200 rpm for 4 days, and culture supernatants were collected by filtration. In parallel, uninoculated PDB and PDB-Trp media were used as controls. The supernatants were lyophilized, the dry weight was measured, and they were kept at 4 °C until hormones extraction. 

Fifty mg (dry weight) of fungal cultures and media supernatant (control) were suspended in 80% methanol-1% acetic acid containing internal standards and mixed by shaking during 60 min at 4 °C. The extract was kept a −20 °C overnight and then centrifuged and the supernatant dried in a vacuum evaporator. The dry residue was dissolved in 1% acetic acid and passed through the Oasis^®^ HLB (reverse phase) column as previously described [61]. 

For GA, IAA, ABA and SA quantification, the dried eluate was dissolved in 5% acetonitrile-1% acetic acid, and the hormones were separated using an autosampler and reverse phase UHPLC chromatography (2.6 µm Accucore RP-MS column, 100 mm length × 2.1 mm i.d., ThermoFisher Scientific) with a 5 to 50% acetonitrile gradient containing 0.05% acetic acid, at 400 µL/min over 21 min. For CK, the extracts were additionally passed through the Oasis^®^ MCX (cationic exchange) and eluted with 60% methanol-5% NH_4_OH to obtain the basic fraction. The final eluate was dried and dissolved in 5% acetonitrile-1% acetic acid and CK were separated with a 5 to 50% acetonitrile gradient over 10 min. The hormones were analyzed with a Q-Exactive mass spectrometer (Orbitrap detector, ThermoFisher Scientific, Waltham, MA, USA) by targeted selected ion monitoring (SIM). The concentrations of hormones in the extracts were determined using embedded calibration curves and the Xcalibur 4.0 and TraceFinder 4.1 SP1 programs. The internal standards for quantification of each of the different plant hormones were the deuterium-labelled hormones. Three independent replicate flasks were analyzed for each strain and culture medium.

### 4.2. Wheat-Trichoderma Greenhouse Assay

The ability of four *Trichoderma* strains, T49, T68, T75 and T115, to promote the growth of wheat plants and induce tolerance to water stress was evaluated in a in vivo assay. Wheat (*Triticum aestivum* L., variety Berdún) seeds were surface disinfected by shaking in 2% sodium hypochlorite for 20 min followed by an additional step of 1 min in 0.1 N HCl, and then rinsed them five times with sterile water. The seeds stratification was conducted for 3 days at 4 °C. *Trichoderma* was applied to the plant growth substrate, and *Trichoderma* cultures were obtained by the inoculation of 0.5 L flasks containing 250 mL of PBD or PDB-Trp medium with 1 × 10^6^ spore/mL and growing of the strains at 28 °C and 180 rpm for 4 days. Then, 250 mL of *Trichoderma* culture (mycelium and supernatant) were used for inoculating 10 pots. 

Surface-disinfected seeds were sown in conical pots (two seeds per pot) of 250 mL capacity containing as substrate a sterile mixture of commercial (Projar Professional, Comercial Projar SA, Fuente el Saz de Jarama, Spain) peat: vermiculite (3:1). The assay initially included 20 treatments and a total of 200 plants, distributed in two blocks (100 plants per block with 10 replicates per treatment) as follows: five for PDB, four PDB cultures and one PDB medium (control); and five for PDB-Trp, four PDB-Trp cultures and one PDB-Trp medium (control). Plants were maintained in a greenhouse at 22 ± 4 °C, as previously described [24], and watered as needed for 2 weeks. Thus, plants from the above indicated two blocks were distributed into 2 sub-blocks as follows: (i) plants from PDB cultures with optimal irrigation; (ii) plants from PDB cultures with water stress (1/3 watering during the third and fourth weeks); (iii) plants from PDB-Trp cultures with optimal irrigation; and (iv) plants from PDB-Trp cultures with water stress. This assay included 10 replicates per condition and lasted 30 days.

#### 4.2.1. Physiological Parameters of Plants

Stomatal conductance (gs) data were taken on 30-day-old wheat plants (10 plants per condition). The gs was measured in the abaxial leaf using a leaf AP4 porometer (Delta-T Devices Ltd., Cambridge, UK). The total shoot of wheat plants was taken at 30 days to record fresh weight (five plants per condition) and dry weight (five plants per conditions), after maintaining plants at 65 °C for 5 days. 

#### 4.2.2. Biochemical Analyses of Plants

Wheat plants of 30 days from the greenhouse assay were used to analyze several enzymatic activities. An intermediate leaf of four wheat plants was collected from each treatment and each considered condition (optimal irrigation and water stress), immediately frozen in liquid nitrogen and ground. Proteins were extracted by homogenizing 50 mg of leaf material in 1 mL of 50 mM potassium phosphate buffer (pH 7.8) and centrifugation at 10,000 rpm for 20 min at 4 °C, and later the supernatant was taken and used for the estimation of activity of superoxide dismutase (SOD), catalase (CAT) and peroxidase (POD) antioxidants enzymes. The activities of CAT and POD were determined by using a spectrophotometer as previously described [62], and one unit defined as the change of 0.01 absorbance unit per min. The activity of SOD was measured according to the previous procedure reported [62] with minor modifications. The mixture reaction contained 2 mL of 50 mM potassium phosphate buffer (pH 7.8), 13 mM methionine, 80 μM nitro blue tetrazolium chloride (NBT), 15 μM riboflavin, and 50 μL of protein extract. One unit of SOD was considered as the amount of enzyme needed to cause 50% inhibition in the photochemical reduction of NBT. The activities of CAT, POD and SOD were expressed as unit per min per mg protein and data were calculated for four biological replicates per considered treatment-condition.

#### 4.2.3. H_2_O_2_ Contents in Wheat Plants

The quantification of H_2_O_2_ was assayed using potassium iodide and by monitoring the absorbance at 390 nm as reported previously [63]. For each sample, fresh plant material was ground in liquid nitrogen and 50 mg used for each sample. Four biological replicates per considered treatment-condition were assayed.

### 4.3. Statistical Analysis

IBM SPSS^®^ Statistics 27 (IBM Corp.) was used for statistical analyses, through an analysis of variance (ANOVA), to test for possible interactions between the main effects (strain, culture medium, stress water) followed by a mean separation using Tukey’s test (*p* < 0.05).

## 5. Conclusions

Four *Trichoderma* strains belonging to genotypically distant species such as *T. virens* T49, *T. longibrachiatum* T68, *T. spirale* T75 and *T. harzianum* T115 were able to produce to a greater or lesser extent not only the already known IAA and SA, but also the CK iP and tZ. However, not all strains produced the phytohormones GA_1_, GA_4_, ABA and the CK DHZ. In addition, the four *Trichoderma* strains displayed ACCD activity. Phytohormone production depended on the strain and/or the composition of the culture medium. *Trichoderma* strains showed different root colonization behavior, with wheat not appearing to be a host for T68. The application of PDB cultures of *Trichoderma* strains can be linked to the ability of wheat plants to adapt the antioxidant machinery and to tolerate water stress. However, non-inoculated PDB-Trp application made water-stressed control plants collapsed, while those treated with *Trichoderma* did not. In any case, the plant’s ROS production and antioxidant activities of none of the treatments with addition of Trp did not seem to respond to water stress, although those corresponding to the application of *Trichoderma* PDB-Trp cultures showed better protection. Plants treated with T49 and T115 showed the best water stress tolerance phenotypes. Perhaps the production of GA_4_ by T49 and ACCD by T115 could be a cause of this good performance of the wheat plants.

## Figures and Tables

**Figure 1 pathogens-10-00991-f001:**
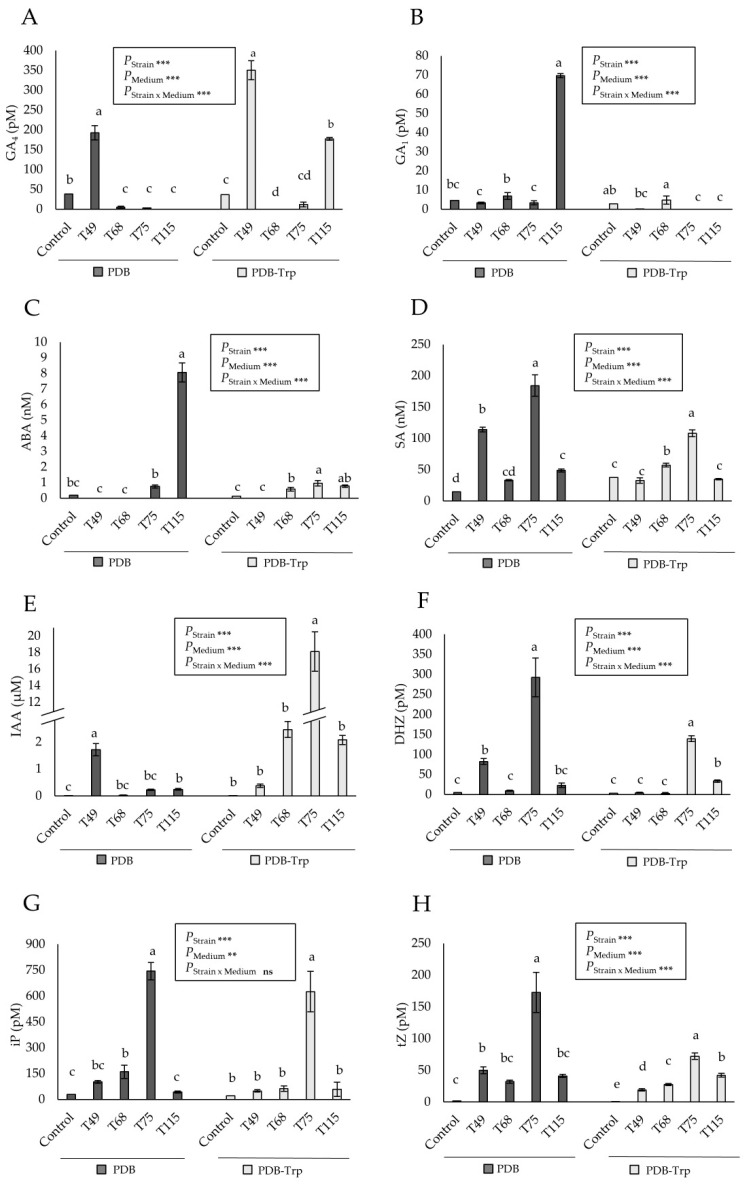
Phytohormone production in 4-days PDB and PDB-tryptophan (Trp) cultures by four *Trichoderma* strains (*T. virens* T49, *T. longibrachiatum* T68, *T spirale* T75 and *T. harzianum* T115) compared to their respective PDB and PDB-Trp media controls. (**A**) Gibberellin 4 (GA_4_), (**B**) gibberellin 1 GA_1_, (**C**) abscisic acid (ABA), (**D**) salicylic acid (SA), (**E**) indole-3-acetic acid (IAA), (**F**) cytokinin dihydrozeatin (DHZ), (**G**) cytokinin isopenteniladenine (iP), and (**H**) cytokinin trans-zeatin (tZ). Data are calculated from *n* = 3 replicates per strain and culture medium. For each phytohormone and culture medium, different letters above the bars indicate significant differences according to one-way analysis of variance (ANOVA) followed by Tukey’s test at the 0.05 alpha-level of confidence. For each phytohormone, significant effects were determined by a two-way ANOVA for *Trichoderma* strain, culture medium and the combination strain per culture medium (***: *p* < 0.001; **: *p* < 0.01; ns: no statistical differences).

**Figure 2 pathogens-10-00991-f002:**
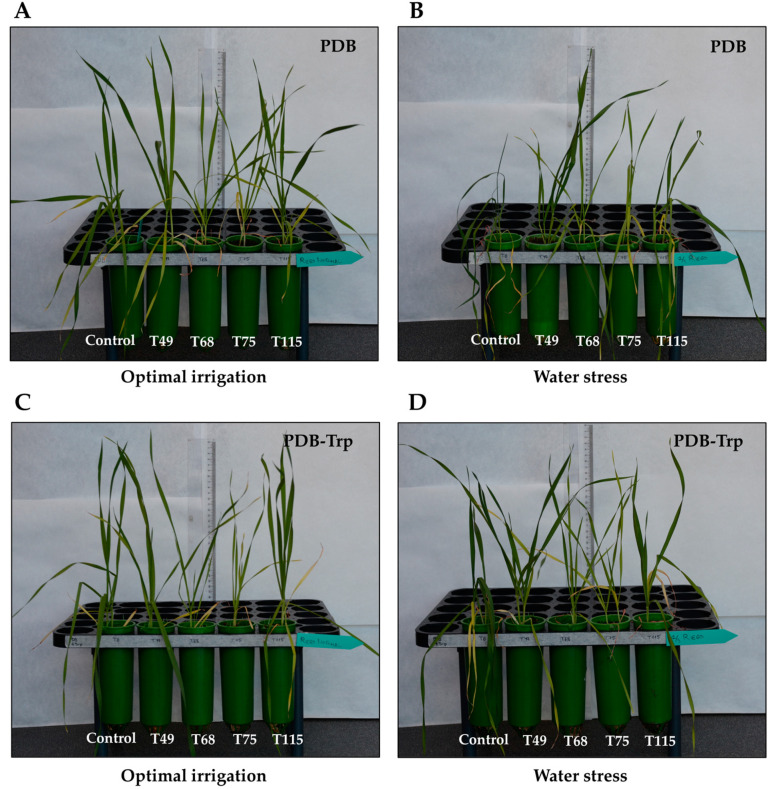
Wheat plants from untreated (control) or treated with *Trichoderma* (*T. virens* T49, *T. longibrachiatum* T68, *T spirale* T75 and *T. harzianum* T115) strains subjected to different irrigation regimes. (**A**) Four-days PDB *Trichoderma* cultures were applied to the plant growth substrate or PDB (control) under optimal irrigation. (**B**) The same under water stress (1/3 of the watering applied during the third and fourth weeks) condition. (**C**) four-days PDB-tryptophan (Trp) *Trichoderma* cultures were applied to the substrate of plant growth or PDB-Trp (control) under optimal irrigation. (**D**) The same under water stress (1/3 of the watering applied during the third and fourth weeks) condition. Photographs were taken when plants were 30 days old.

**Figure 3 pathogens-10-00991-f003:**
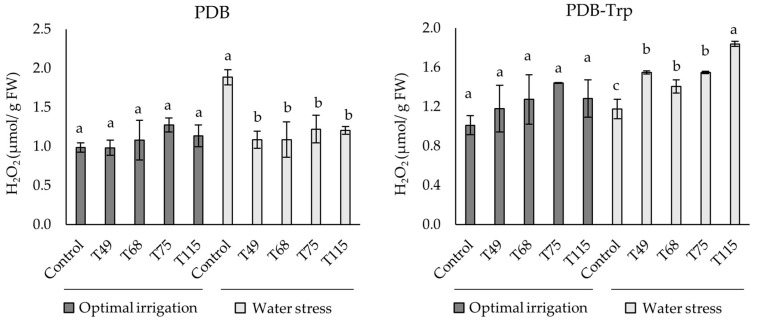
Effect of *T. virens* T49, *T. longibrachiatum* T68, *T spirale* T75 and *T. harzianum* T115 treatments from 4-days PDB and PDB-tryptophan (Trp) cultures on H_2_O_2_ content in wheat plant leaf grown under optimal irrigation and water stress (1/3 of the watering applied during the third and fourth weeks) conditions. Values are expressed in μmoles of H_2_O_2_ per g of leaf fresh weight (FW). Data are calculated from *n* = 4 replicates for each strain, culture medium and plant growth condition. For each fungal culture medium and plant growth conditions, different letters above the bars indicate significant differences according to one-way analysis of variance (ANOVA) followed by Tukey’s test at the 0.05 alpha-level of confidence.

**Figure 4 pathogens-10-00991-f004:**
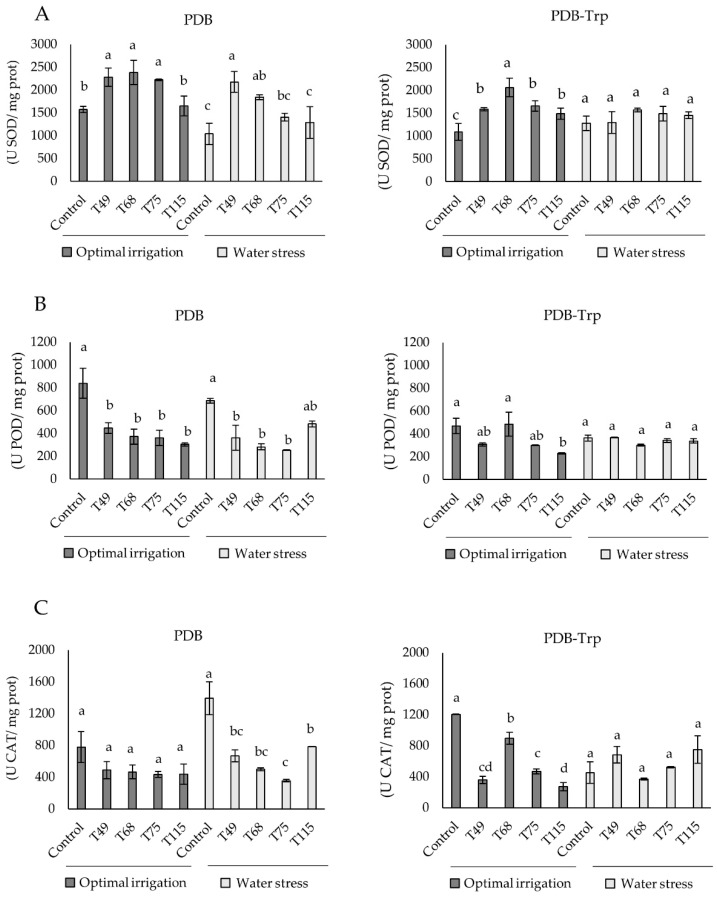
Effect of *Trichoderma* strain (*T. virens* T49, *T. longibrachiatum* T68, *T spirale* T75 and *T. harzianum* T115) treatments from 4-days PDB and PDB-tryptophan (Trp) cultures on (**A**) SOD, (**B**) POD and (**C**) CAT activities of wheat plants grown under optimal irrigation and water stress (1/3 of the watering applied during the third and fourth weeks) conditions. Data are calculated from *n* = 4 replicates for each strain, culture medium and plant growth condition. For each fungal culture medium and plant growth condition, different letters above the bars indicate significant differences according to one-way analysis of variance (ANOVA) followed by Tukey’s test at the 0.05 alpha-level of confidence. SOD: superoxide dismutase, POD: peroxidase, and CAT: catalase.

**Table 1 pathogens-10-00991-t001:** Source, origin and accession numbers of *Trichoderma* strains included in this study.

Strain	Identified as	Source	Origin	GenBank NumbersITS//tef1α
T49	*T. virens*	soil	Brazil	MZ312097//MZ346026
T68	*T. longibrachiatum*	soil	Brazil	MZ311298//MZ346027
T75	*T. spirale*	soil	Spain	MZ311299//MZ346028
T115	*T. harzianum*	soil	Philippines	MZ313912//MZ346029

**Table 2 pathogens-10-00991-t002:** Colony growth of *Trichoderma* strains, expressed in cm, on PDA, PDA-Trp and MEA after 48 h at 28 °C, and sporulation rate (spore/mL) measured at 10 days of incubation.

Strain	Growth Rate			Spores Produced		
	PDA	PDA-Trp	MEA	PDA	PDA-Trp	MEA
T49	7.1 a	6.9 a	7.0 a	2.5 × 10^8^ a	4.1 × 10^8^ b	1.46 × 10^8^ a
T68	7.0 a	6.3 b	4.7 c	2.4 × 10^8^ a	4.2 × 10^8^ b	6.2 × 10^7^ b
T75	4.1 c	4.0 d	3.6 d	6.8 × 10^6^ b	7.7 × 10^5^ c	1.2 × 10^7^ b
T115	5.4 b	5.4 c	5.4 b	4.0 × 10^8^ a	7.7 × 10^8^ a	4.9 × 10^7^ b

Data are calculated from *n* = 4 replicates per condition. Values in the same column with different letters are significantly different according to one-way analysis of variance (ANOVA) followed by Tukey’s test at the 0.05 alpha-level of confidence.

**Table 3 pathogens-10-00991-t003:** Colonization of wheat roots by *Trichoderma* strains (*T. virens* T49, *T. longibrachiatum* T68, *T spirale* T75 and *T. harzianum* T115) *.

Strains	*Trichoderma* Actin				Wheat Ctin				Ratio ****
	Ct	SD	Qty **	SD	Ct	SD	Qty ***	SD	
T49	18.06	0.07	3.08	0.80	22.85	0.11	2.47	0.73	1.39 ± 0.71 a
T68	19.40	0.04	0.16	0.18	22.36	0.30	2.99	1.14	0.04 ± 0.04 c
T75	17.39	0.02	5.13	0.72	21.95	0.09	4.70	0.14	1.09 ± 0.13 ab
T115	18.66	0.10	2.74	0.54	21.89	0.15	4.22	0.45	0.65 ± 0.11 b

* Fungal DNA present on wheat roots 42 h after the inoculation was quantified by qPCR. Ct, threshold cycle and SD, standard deviation. ** Quantity of *Trichoderma* DNA (ng) referred to *Trichoderma* actin gene. *** Quantity of wheat DNA (ng) referred to wheat actin gene. **** Proportion of fungal DNA vs. plant DNA. Data are calculated from *n* = 4 replicates per strain. Values in the same column with different letters are significantly different according to one-way analysis of variance (ANOVA) followed by Tukey’s test at the 0.05 alpha-level of confidence.

**Table 4 pathogens-10-00991-t004:** Effect of 4-days PDB cultures of *Trichoderma* strains on mean fresh and dry weight and conductance values of 30-day-old wheat plants grown in greenhouse with optimal irrigation and water stress conditions (1/3 in the last two weeks).

Treatment	Fresh Weight (g)		Dry Weight (g)		gs (mol H_2_O m^−2^ s^−1^)	
	Optimal Irrigation	Water Stress	Optimal Irrigation	Water Stress	Optimal Irrigation	Water Stress
Control	1.13 a	0.39 b	0.22 a	0.10 b	0.166 b	0.006 b
T49	1.09 a	0.82 a	0.21 a	0.22 a	0.278 a	0.089 a
T68	0.51 b	0.35 b	0.11 b	0.07 b	0.108 b	0.110 a
T75	0.35 b	0.41 b	0.10 b	0.08 b	0.139 b	0.132 a
T115	0.93 a	0.76 a	0.20 a	0.17 a	0.234 a	0.100 a

Data are calculated from *n* = 10 replicates per treatment and condition. Values in the same column with different letters are significantly different according to one-way analysis of variance (ANOVA) followed by Tukey’s test at the 0.05 alpha-level of confidence.

**Table 5 pathogens-10-00991-t005:** Effect of 4-days PDB-Trp cultures of *Trichoderma* strains on mean fresh and dry weight and conductance values of 30-day-old wheat plants grown in greenhouse with optimal irrigation and water stress conditions (1/3 in the last two weeks).

Treatment	Fresh Weight (g)		Dry Weight (g)		gs (mol H_2_O m^−2^ s^−1^)	
	Optimal Irrigation	Water Stress	Optimal Irrigation	Water Stress	Optimal Irrigation	Water Stress
Control	1.26 a	0.82 a	0.20 a	0.19 a	0.189 ab	0.157 a
T49	1.32 a	0.67 a	0.23 a	0.22 a	0.213 a	0.188 a
T68	0.68 b	0.52 a	0.12 b	0.11 b	0.147 b	0.165 a
T75	0.55 b	0.61 a	0.11 b	0.12 b	0.150 b	0.161 a
T115	1.27 a	0.74 a	0.24 a	0.21 a	0.162 ab	0.108 a

Data are calculated from *n* = 10 replicates per treatment and condition. Different letters indicate significant differences within each column according to one-way analysis of variance (ANOVA) followed by Tukey’s test at the 0.05 alpha-level of confidence.

## Data Availability

Not applicable.

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
