# Peer review of "Phytohormone Production Profiles in Trichoderma Species and Their Relationship to Wheat Plant Responses to Water Stress"

_pathogens, 2021, doi:10.3390/pathogens10080991_

Round 1

Reviewer 1 Report

The manuscript deals with an interesting and important subject; the phytohormone production of Trichoderma species and their effect on wheat production. The authors used modern experimental techniques and provided many new data with reliable statistics. The only thing I miss that the authors should emphasize the highlights, the real new findings in their results. I know that plant hormones form a network, and any change in the content of one hormone effects the others.

In addition, the growth promoting (and plant protection) effect of Trichodermas is complex as well. It depends not only on production of plant hormones.

In spite of the above arguments I would like to see suggestions from the authors on the possible practical application of certain Trichoderma species among certain circumstances to improve wheat production.

Furthermore, from the rather inhibitory effect of cultures on non-stressed control plants, it looks like that the treatment itself is a stress to wheat. Could the author comment on it?

Author Response

Dear Editor,

Thank you very much for your comments and those of the reviewers, which have certainly helped to improve the quality and information facilitated in our manuscript. Changes are highlighted in blue shading. We have included a new Table 3 with the results of root colonization by the Trichoderma strains, therefore the previous Table 3 is now the new Table 4. Similarly, the previous Table 4 is now the new Table 5. The Discussion has also been modified following the indications of Reviewer 1 for “highlighting the real new findings” and “the possible practical application of Trichoderma strains” and shortened in other parts to comply with Reviewer 2's mandate. Here are our responses to the issues raised by the reviewers.

Report 1

Comments and Suggestions for Authors

The manuscript deals with an interesting and important subject; the phytohormone production of Trichoderma species and their effect on wheat production. The authors used modern experimental techniques and provided many new data with reliable statistics. The only thing I miss that the authors should emphasize the highlights, the real new findings in their results. I know that plant hormones form a network, and any change in the content of one hormone effects the others. In addition, the growth promoting (and plant protection) effect of Trichoderma is complex as well. It depends not only on production of plant hormones.

In spite of the above arguments, I would like to see suggestions from the authors on the possible practical application of certain Trichoderma species among certain circumstances to improve wheat production.

Answer:

We have emphasized our findings and the utility of Trichoderma strains by including the following text in the last paragraph of the Discussion (lines 379-390):

“The production of the phytohormones GAs, ABA, SA, IAA and CKs by Trichoderma species is a strain-specific characteristic and depends on the composition of the culture medium. These differences are a factor to be considered when exploring the beneficial effects of Trichoderma on plants. In this way, the T. virens T49 and T. harzianum T115 cultures were the best performers in alleviating wheat plants from water stress and it was precisely these two strains which exhibited GA1 and IAA, and GA4 and ABA production, respectively, in media not supplemented with Trp. The present work contributes to highlighting the role that the balance of phytohormone levels, to which Trichoderma contributes with its own production, plays in beneficial plant-Trichoderma interactions. In any case, the growth promotion and plant protection effects of Trichoderma are mechanisms with complex regulation that depends on other Trichoderma traits and not only on the production of phytohormones by this fungus. The results of this work are an example of the usefulness of Trichoderma strains in the protection of crop plants against abiotic stresses”.

Furthermore, from the rather inhibitory effect of cultures on non-stressed control plants, it looks like that the treatment itself is a stress to wheat. Could the author comment on it?

Answer:

Trichoderma acts in the plant on the nodes that regulate defence and growth. This means that plants with activated defences grow less and vice versa. We have included this new text in the Discussion (this new text refers to the new Table 3 produced with the root colonization results suggested by Reviewer 2):

Lines 341-343: “Since strain T68 was unable to colonize the wheat root, it may be releasing some other metabolites that could limit the growth of the plant”.

Lines 346-348: “The importance of selecting a suitable strain of Trichoderma is a key point in this type of studies as it has been observed that the colonization of Arabidopsis, tomato and maize roots by T. virens Gv29.8 led to reduced growth of both roots and stems [7,51,52]”.

Reviewer 2 Report

Dear authors,

The manuscript “Phytohormone production profiles in Trichoderma species and their
relationship to wheat plant responses to water stress” is “in general” well written and well analyzed. Minor spell checking is needed.

Here are some concerns or questions:

-Differentiation between strains and species. The authors used individual strains of 4 different species, but mainly use the term strains in an equal manner to species in the manuscript throughout. Please make a clear distinction.

-In the introduction the rational for testing water stress (drought in this case) and the use of Trichoderma species is missing (please include references showing the connection of Trichoderma and drought). Furthermore, please also include the reasoning/connection of drought stress and ROS.  

- Table 3: Is wheat for all the species listed a host? For T68 and for T75 even under optimal water conditions the plant growth is generally negatively affected. They almost seem to have a pathogenic behavior. This leads to the question of how is wheat colonized/how does it interact with the different strains of those species under the tested conditions? Please include microscopy or other evidence of root colonisation/ interactions.

-Table 4: PDB-Trp alone is actually as good as the best interaction of wheat with Trichoderma in alleviating drought, even if compared to PDB alone values in Table 3. Thus is the drought stress alleviated by Trp alone? Please include a plant drought growth test which tests Trp in water alone. If that is the case Trp could be used to prime, stimulate resistance against drought in the field and Trichoderma would not even be needed.

-Figure 3: Is the unit really mM H2O2? 4-8mM H2O2 per g fresh weight is not normal for plants and would be toxic, it should be in the um range. Please check the units. Better check units for other graphs too, to make sure they are correct.

-Figure 4: It would be better to use the same scale of the axis for PDB and PDB-Trp. For example, the SOD in the control plants with optimal irrigation shows that PDB-Trp is only 2/3 of the PDB samples. It would make it easier to compare. Further, the section discussing those values (page7 line 227 and following) is difficult to understand and I’m not sure it reflects the results from figure 4 perfectly. Please check and clarify.              

-The discussion seems to be a bit too long. Please check and see if there are ways to shorten it and to be more concise.   

Table 2: mark a clear distinction between growth rate and sporulation on the different media. Maybe line 2 and 3 from the top should have a gap to separate those 2?

Page 5 line 151: strain that (should be than)

Page 9 line 274: strain T68 showed good grew (growth) and sporulated (sporulation)

Author Response

Dear Editor,

Thank you very much for your comments and those of the reviewers, which have certainly helped to improve the quality and information facilitated in our manuscript. Changes are highlighted in blue shading. We have included a new Table 3 with the results of root colonization by the Trichoderma strains, therefore the previous Table 3 is now the new Table 4. Similarly, the previous Table 4 is now the new Table 5. The Discussion has also been modified following the indications of Reviewer 1 for “highlighting the real new findings” and “the possible practical application of Trichoderma strains” and shortened in other parts to comply with Reviewer 2's mandate. Here are our responses to the issues raised by the reviewers.

Report 2

Dear authors,

The manuscript “Phytohormone production profiles in Trichoderma species and their
relationship to wheat plant responses to water stress” is “in general” well written and well analyzed. Minor spell checking is needed.

Here are some concerns or questions:

- Differentiation between strains and species. The authors used individual strains of 4 different species, but mainly use the term strains in an equal manner to species in the manuscript throughout. Please make a clear distinction.

Answer:

The reviewer is right. We have worked with representative strains of phylogenetically distant species and for this reason we have referred to the term "strain" throughout the text. When we refer to a trait, an activity, or the production of a phytohormone, we must say that it corresponds to a strain, as we do not know if the rest of the strains of that species behave in the same way. However, there are phrases that address a broader context and where we should say "species". We have added this new text to facilitate the understanding of four sentences that were not clear in the previous version:

L45-46: Selected Trichoderma species also produce effector molecules…

L48-49: In this regard, rhizosphere competent species have evolved to manipulate root development, plant immunity and stress tolerance…

L100-101: “…four Trichoderma strains of four different species representing the genetic diversity of the genus.”

Line 530 (Conclusions): “Four Trichoderma strains belonging to genotypically distant species such as…”

- In the introduction the rational for testing water stress (drought in this case) and the use of Trichoderma species is missing (please include references showing the connection of Trichoderma and drought). Furthermore, please also include the reasoning/connection of drought stress and ROS.

Answer:

We have included the following text in the Introduction:

Lines 58-65: “T. afroharzianum (formerly T. harzianum) T22 improved tolerance of tomato seedlings to water deficit [25]. The colonization of cacao seedlings by T. hamatum DIS 219b enhanced seedling growth, altered gene expression, and delayed the onset of the cacao drought response in leaves [26]. Similarly, T. atroviride ID20G inoculation of seeds ameliorated drought stress-induced damages by improving antioxidant defence in maize seedlings [27]. The same happened with the improved drought tolerance observed in rice genotypes inoculated with T. harzianum Th-56 in which the antioxidant machinery was activated in a dose-dependent manner [28]”.

Lines 89-92: “ROS are key players in the complex signaling network of plant responses to drought stress, so it is essential to maintain ROS at non-toxic levels in a delicate balancing act between ROS production, involving ROS generating enzymes and the unavoidable production of ROS during basic cellular metabolism, and ROS-scavenging pathways [39]”.

Lines 94-99: “There is sufficient evidence to consider that Trichoderma association can help plants in sustaining drought stress by increasing: i) the expression of antioxidative enzymes that alleviate the damage caused by the accumulation of ROS and modulating the balance of plant’s phytohormones [25,41]; ii) the absorption surface that leads the plant to improve water-use efficiency [33]; and iii) the synthesis of phytohormones and phytohormonal analogues to promote plant performance”.

- Table 3: Is wheat for all the species listed a host? For T68 and for T75 even under optimal water conditions the plant growth is generally negatively affected. They almost seem to have a pathogenic behavior. This leads to the question of how is wheat colonized/how does it interact with the different strains of those species under the tested conditions? Please include microscopy or other evidence of root colonisation/ interactions.

Answer:

We no longer have any of this material (wheat roots from our greenhouse assay), but a new experiment has been set up with 10-day-old wheat seedlings that we had available and with which we were able to carry out colonization test to analyze the ability of the Trichoderma strains for wheat root colonization (new Table 3).

We have also included the following new text:

Lines 18-19 (abstract): “These four strains showed different degrees of wheat root colonization”.

Lines 100-102: “we have used four Trichoderma strains of four different species representing the genetic diversity of the genus, in which we analyzed their capacity for wheat root colonization, …”.

Lines 129-134: 2.2. Differences in colonization of roots of wheat seedlings by Trichoderma strains. In order to perform a comparative analysis of the wheat root colonization ability among the four Trichoderma strains, we determined the proportion of fungal DNA vs. plant DNA from qPCR data in 10-day-old seedling roots at 42 h after fungal inoculation. As shown in Table 3, strains T49, T75 and T115 colonized the roots, with the highest rates for T49 and T75 (p < 0.05), while T68 showed no colonization.

Lines 285-287: “Root colonization ability is often a criterion for selecting Trichoderma strains beneficial to plants [12], and we found that wheat was not a host for strain T68”

Lines 418-437 (M&M): New 4.1.3. Root colonization assay

- Table 4 (new Table 5): PDB-Trp alone is actually as good as the best interaction of wheat with Trichoderma in alleviating drought, even if compared to PDB alone values in Table 3. Thus is the drought stress alleviated by Trp alone? Please include a plant drought growth test which tests Trp in water alone. If that is the case Trp could be used to prime, stimulate resistance against drought in the field and Trichoderma would not even be needed.

Answer:

The answer is: No. Two aspects should not be overlooked: i) standard deviations are often very large when working with plant sizes at greenhouse level, and ii) plants of the Trp control treatment compared to those from the non-Trp treatment may appear to maintain their size, but they look like they are collapsing because they cannot withstand the drought stress (Figure 2). The collapse is very evident when the plants are observed in vivo.

- Figure 3: Is the unit really mM H2O2? 4-8mM H2O2 per g fresh weight is not normal for plants and would be toxic, it should be in the um range. Please check the units. Better check units for other graphs too, to make sure they are correct.

Answer:

Sorry, it was a mistake. Many thanks. A new Figure 3 with a suitable scale has been produced

- Figure 4: It would be better to use the same scale of the axis for PDB and PDB-Trp. For example, the SOD in the control plants with optimal irrigation shows that PDB-Trp is only 2/3 of the PDB samples. It would make it easier to compare. Further, the section discussing those values (page7 line 227 and following) is difficult to understand and I’m not sure it reflects the results from figure 4 perfectly. Please check and clarify.  

Answer:

Done. Scales have been adapted in a new Figure 4.            

- The discussion seems to be a bit too long. Please check and see if there are ways to shorten it and to be more concise.  

Answer:

Following your advice, the Discussion has been shortened. However, Reviewer 1 has suggested to include a less abrupt ending than the one we had, so we have tried to follow the advice of both of you. In any case, the text withdrawn (former lines 258-260, 271-273, 298-302) outweighs the text added.

Table 2: mark a clear distinction between growth rate and sporulation on the different media. Maybe line 2 and 3 from the top should have a gap to separate those 2?

Answer: done

Page 5 line 151: strain that (should be than).

Answer: done

Page 9 line 274: strain T68 showed good grew (growth) and sporulated (sporulation)

Answer: done.

Round 2

Reviewer 2 Report

Dear authors,

The manuscript clearly has improved and I have no further concerns.